# Decoupling Global and Local Representations via Invertible Generative Flows

**Xuezhe Ma**[1][*]**, Xiang Kong**[2]**, Shanghang Zhang**[3]**, Eduard Hovy**[2]
[1]University of Southern California
[2]Carnegie Mellon University
[3]University of California, Berkeley
xuezhema@isi.edu, xiangk@cs.cmu.edu, shz@berkeley.edu

## Abstract

In this work, we propose a new generative model that is capable of automatically decoupling global and local representations of images in an entirely unsupervised setting, by embedding a generative flow in the VAE framework to model the decoder. Specifically, the proposed model utilizes the variational auto-encoding framework to learn a (low-dimensional) vector of latent variables to capture the global information of an image, which is fed as a conditional input to a flow-based invertible decoder with architecture borrowed from style transfer literature. Experimental results on standard image benchmarks demonstrate the effectiveness of our model in terms of density estimation, image generation and unsupervised representation learning. Importantly, this work demonstrates that with only architectural inductive biases, a generative model with a likelihood-based objective is capable of learning decoupled representations, requiring no explicit supervision. The code for our model is available at https://github.com/XuezheMax/wolf.

## 1 Introduction

Unsupervised learning of probabilistic models and meaningful representation learning are two central yet challenging problems in machine learning. Formally, let $X \in \mathcal{X}$ be the random variables of the observed data, e.g., $X$ is an image. One goal of generative models is to learn the parameter $\theta$ such that the model distribution $P_\theta(X)$ can best approximate the true distribution $P(X)$. Throughout the paper, uppercase letters represent random variables and lowercase letters their realizations.

Unsupervised (disentangled) representation learning, besides data distribution estimation and data generation, is also a principal component in generative models. The goal is to identify and disentangle the underlying causal factors, to tease apart the underlying dependencies of the data, so that it becomes easier to understand, to classify, or to perform other tasks (Bengio et al., 2013). Unsupervised representation learning has spawned significant interests and a number of techniques (Chen et al., 2017a; Devlin et al., 2019; Hjelm et al., 2019) has emerged over the years to address this challenge. Among these generative models, VAE (Kingma & Welling, 2014; Rezende et al., 2014) and Generative (Normalizing) Flows (Dinh et al., 2014) have stood out for their simplicity and effectiveness.

### 1.1 Variational Auto-Encoders (VAEs)

VAE, as a member of latent variable models (LVMs), gains popularity for its capability of automatically learning meaningful (low-dimensional) representations from raw data. In the framework of VAEs, a set of latent variables $Z \in \mathcal{Z}$ are introduced, and the model distribution $P_\theta(X)$ is defined as the marginal of the joint distribution between $X$ and $Z$:

$$p_\theta(x) = \int_{\mathcal{Z}} p_\theta(x, z) d\mu(z) = \int_{\mathcal{Z}} p_\theta(x|z) p_\theta(z) d\mu(z), \quad \forall x \in \mathcal{X}, \tag{1}$$

where the joint distribution $p_\theta(x, z)$ is factorized as the product of a prior $p_\theta(z)$ over the latent $Z$, and the "generative" distribution $p_\theta(x|z)$. $\mu(z)$ is the base measure on the latent space $\mathcal{Z}$.

---

[*]Work was done at Carnegie Mellon University.

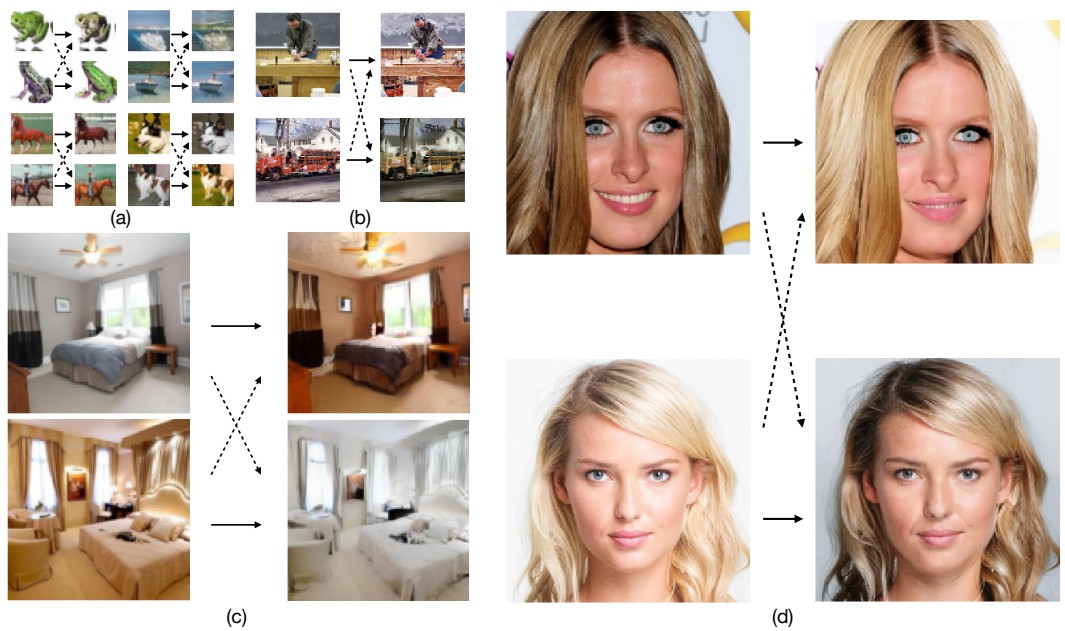

Figure 1: Examples of the switch operation, which switches the global representations of two images from four datasets: (a) CIFAR-10, (b) ImageNet, (c) LSUN Bedroom and (d) CelebA-HQ.

In general, this marginal likelihood is intractable to compute or differentiate directly, and Variational Inference (Wainwright et al., 2008) provides a solution to optimize the *evidence lower bound* (ELBO), an alternative objective by introducing a parametric *inference model* $q_\phi(z|x)$:

$$\mathrm{E}_{p(X)}\left[\log p_\theta(X)\right] \geq \mathrm{E}_{p(X)}\left[\mathrm{E}_{q_\phi(Z|X)}[\log p_\theta(X|Z)] - \mathrm{KL}(q_\phi(Z|X)||p_\theta(Z))\right] \quad (2)$$

where ELBO could be seen as an autoencoding loss with $q_\phi(z|x)$ being the encoder and $p_\theta(x|z)$ being the decoder, with the first term in the RHS in (2) as the reconstruction error.

## 1.2 GENERATIVE FLOWS

Put simply, generative flows (a.k.a., normalizing flows) work by transforming a simple distribution, $P(\Upsilon)$ (e.g. a simple Gaussian) into a complex one (e.g. the complex distribution of data $P(X)$) through a chain of invertible transformations.

Formally, a generative flow defines a bijection function $f : \mathcal{X} \to \Upsilon$ (with $g = f^{-1}$), where $\upsilon \in \Upsilon$ is a set of latent variables with simple prior distribution $p_\Upsilon(\upsilon)$. It provides us with an invertible transformation between $X$ and $\Upsilon$, whereby the generative process over $X$ is defined straightforwardly:

$$\upsilon \sim p_\Upsilon(\upsilon), \quad \text{then } x = g_\theta(\upsilon). \quad (3)$$

An important insight behind generative flows is that given this bijection function, the change of the variable formula defines the model distribution on $X$ by:

$$p_\theta(x) = p_\Upsilon\left(f_\theta(x)\right)\left|\det\left(\frac{\partial f_\theta(x)}{\partial x}\right)\right|, \quad (4)$$

where $\frac{\partial f_\theta(x)}{\partial x}$ is the Jacobian of $f_\theta$ at $x$. A stacked sequence of such invertible transformations is called a generative (normalizing) flow (Rezende & Mohamed, 2015):

$$X \underset{g_1}{\overset{f_1}{\longleftrightarrow}} H_1 \underset{g_2}{\overset{f_2}{\longleftrightarrow}} H2 \underset{g_3}{\overset{f_3}{\longleftrightarrow}} \cdots \underset{g_K}{\overset{f_K}{\longleftrightarrow}} \Upsilon,$$

where $f = f_1 \circ f_2 \circ \cdots \circ f_K$ is a flow of $K$ transformations (omitting $\theta$ for brevity).

### 1.3 PROBLEMS OF VAEs AND GENERATIVE FLOWS

Despite their impressive successes, VAEs and generative flows still suffer their own problems.

**Posterior Collapse in VAEs** As discussed in Bowman et al. (2015), without further assumptions, the ELBO objective in (2) may not guide the model towards the intended role for the latent variables $Z$, or even learn uninformative $Z$ with the observation that the KL term $\mathrm{KL}(q_\phi(Z|X)||p_\theta(Z))$ vanishes to zero. The essential reason of this *posterior collapse* problem is that, under absolutely unsupervised setting, the marginal likelihood-based objective incorporates no (direct) supervision on the latent space to characterize the latent variable $Z$ with preferred properties w.r.t. representation learning.

**Local Dependency in Generative Flows** Generative flows suffer from the limitation of expressiveness and local dependency. Most generative flows tend to capture the dependency among features only locally, and are incapable of realistic synthesis of large images compared to GANs (Goodfellow et al., 2014). Unlike latent variable models, e.g. VAEs, which represent the high-dimensional data as coordinates in a latent low-dimensional space, the long-term dependencies that usually describe the global features of the data can only be propagated through a composition of transformations. Previous studies attempted to enlarge the receptive field by using a special design of parameterization like masked convolutions (Ma et al., 2019a) or attention mechanism (Ho et al., 2019).

In this paper, we propose a simple and effective generative model to simultaneously tackle the aforementioned challenges of VAEs and generative flows by leveraging their properties to complement each other. By embedding a generative flow in the VAE framework to model the decoder, the proposed model is able to learn decoupled representations which capture global and local information of images respectively in an entirely unsupervised manner. The key insight is to utilize the inductive biases from the model architecture design — leveraging the VAE framework equipped with a compression encoder to extract the global information in a low-dimensional representation, and a flow-based decoder which favors local dependencies to store the residual information into a local high-dimensional representation (§2). Experimentally, on four benchmark datasets for images, we demonstrate the effectiveness of our model on two aspects: (i) density estimation and image generation, by consistently achieving significant improvements over Glow (Kingma & Dhariwal, 2018), (ii) decoupled representation learning, by performing classification on learned representations the *switch operation* (see examples in Figure 1). Perhaps most strikingly, we demonstrate the feasibility of decoupled representation learning via the plain likelihood-based generation, using only architectural inductive biases (§3).

## 2 GENERATIVE MODEL FOR DECOUPLED REPRESENTATION LEARNING

We first illustrate the high-level insights of the architecture design of our generative model (shown in Figure 2) before detailing each component in the following sections.

### 2.1 GENERATIVE MODEL ARCHITECTURE

In the training process of our generative model, we minimize the negative ELBO in VAE:

$$\mathcal{L}_{elbo}(\theta, \phi) = \mathrm{E}_{p(X)}\left[\mathrm{E}_{q_\phi(Z|X)}[-\log p_\theta(X|Z)] + \mathrm{KL}(q_\phi(Z|X)||p_\theta(Z))\right] \tag{5}$$

where $\mathcal{L}_{elbo}(\theta, \phi)$ is the negative ELBO of RHS in (2). Specifically, we first feed the input image $x$ into the encoder $q_\phi(z|x)$ to compute the latent variable $z$. The encoder is designed to be a compression network, which compresses the high-dimensional image into a low-dimensional vector (§2.2). Through this compression process, the local information of an image $x$ is enforced to be discarded, yielding representation $z$ that captures the global information. Then we feed $z$ as a conditional input to a flow-based decoder, which transforms $x$ into the representation $v$ with the same dimension (§2.3). Since the decoder is invertible, with $z$ and $v$, we can exactly reconstruct the original image $x$. It indicates that $z$ and $v$ maintain all the information of $x$, and the reconstruction process can be regarded as an additional operation — adding $z$ and $v$ to recover $x$. In this way, we expect that the local information discarded in the compression process will be restored in $v$.

In the generative process, we combine the sampling procedures of VAEs and generative flows: we first sample a value of $z$ from the prior distribution $p(z)$, and a value of $v$ from $p_\Upsilon(v)$; second, we input $z$ and $v$ into the invertible function $f^{-1}$ modeled by the generative flow decoder to generate an image $x = f_\theta^{-1}(v, z) = g_\theta(v, z)$.

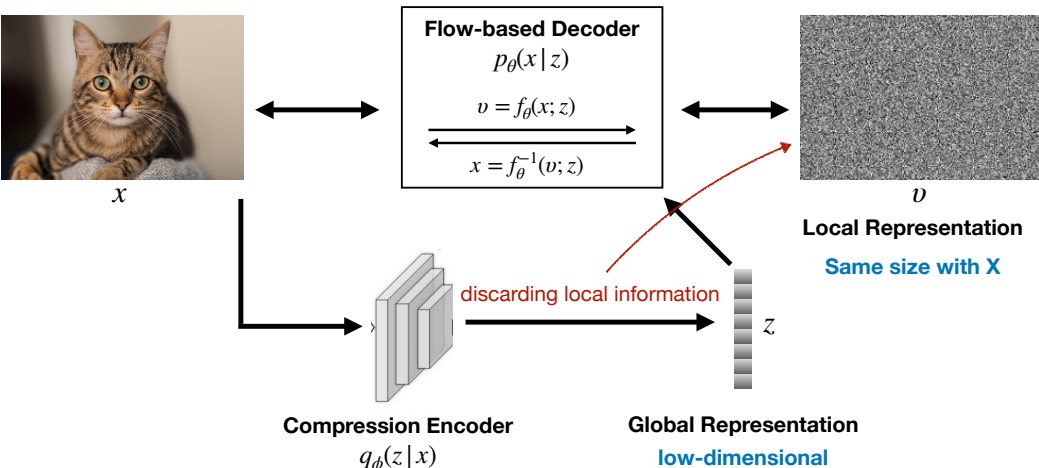

Figure 2: Diagram to illustrate the process of decoupling an image $x$ into the global representation $z$ and local representation $\upsilon$. The key insight is the architecture design of the compression encoder and the invertible decoder.

## 2.2 COMPRESSION ENCODER

Following previous work, the variational posterior distribution $q_\phi(z|x)$, a.k.a encoder, models the latent variable $Z$ as a diagonal Gaussian with learned mean and variance:

$$q_\phi(z|x) = \mathcal{N}(z; \mu(x), \sigma^2(x)) \tag{6}$$

where $\mu(\cdot)$ and $\sigma(\cdot)$ are neural networks. In the context of 2D images where $x$ is a tensor of shape $[h \times w \times c]$ with spatial dimensions $(h, w)$ and channel dimension $c$, the compression encoder maps each image $x$ to a $d_z$-dimensional vector. $d_z$ is the dimension of the latent space.

In this work, the motivation of the encoder is to compress the high-dimensional data $x$ to low-dimensional latent variable $z$, i.e. $h \times w \times c \gg d_z$, to enforce the latent representation $z$ to capture the global features of $x$. Furthermore, unlike previous studies on VAE based generative models for natural images (Kingma et al., 2016; Chen et al., 2017a; Ma et al., 2019b) that represented latent codes $z$ as low-resolution feature maps[1], we represent $z$ as an unstructured 1-dimensional vector to erase the local spatial dependencies. Concretely, we implement the encoder with a similar architecture in ResNet (He et al., 2016). The spatial downsampling is implemented by a 2-strided ResNet block with $3 \times 3$ filters. On top of these ResNet blocks, there is one more fully-connected layer with number of output units equal to $d_z \times 2$ to generate $\mu(x)$ and $\log \sigma^2(x)$ (details in Appendix B).

**Zero initialization.** Following Ma et al. (2019c), we initialize the weights of the last fully-connected layer that generates the $\mu$ and $\log \sigma^2$ values with zeros. This ensures that the posterior distribution is initialized as a simple normal distribution, which has been demonstrated helpful for training very deep neural networks more stably in the framework of VAEs.

## 2.3 INVERTIBLE DECODER BASED ON GENERATIVE FLOW

The flow-based decoder defines a (conditionally) invertible function $\upsilon = f_\theta(x; z)$, where $\upsilon$ follows a standard normal distribution $\upsilon \sim \mathcal{N}(0, I)$. Conditioned on the latent variable $z$ output from the encoder, we can reconstruct $x$ with the inverse function $x = f_\theta^{-1}(\upsilon; z)$. The flow-based decoder adopts the main backbone architecture of Glow (Kingma & Dhariwal, 2018), where each step of flow consists of the same three types of elementary flows — actnorm, invertible $1 \times 1$ convolution and coupling (details in Appendix A).

---

[1]For example, the latent codes of the images from CIFAR-10 corpus with size $32 \times 32$ are represented by 16 feature maps of size $8 \times 8$ in Kingma et al. (2016); Chen et al. (2017a).

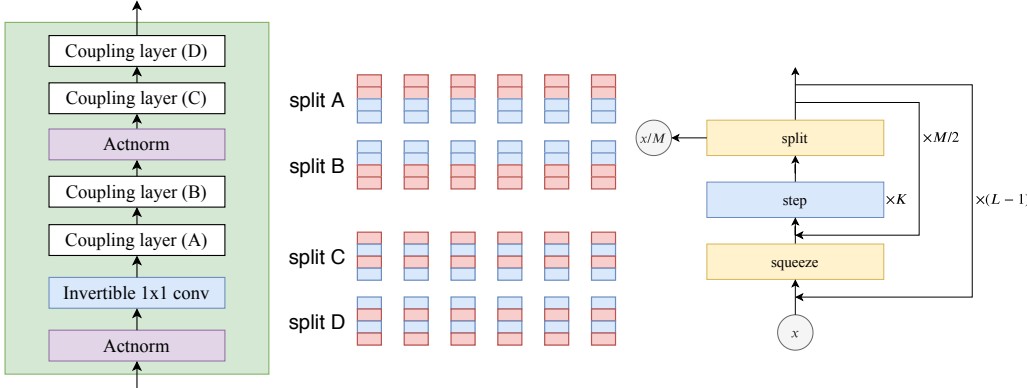

(a) One step of our flow     (b) Four types of coupling layer splits   (c) Fine-grained multi-scale architecture

Figure 3: The refined architecture of Glow that used in our decoder. (a) The architecture of one re-organized step. (b) The visualization of four split patterns for coupling layers, where the red color denotes $x_a$ and the blue color denotes $x_b$. (c) The fine-grained version of multi-scale architecture.

**Conditional Inputs in Affine Coupling Layers.**    To incorporate $z$ as a conditional input to the decoder, we modify the neural networks for the scale and bias terms:

$$
\begin{aligned}
x_a, x_b &= \text{split}(x) \\
y_a &= x_a \\
y_b &= \text{s}(x_a, z) \odot x_b + \text{b}(x_a, z) \\
y &= \text{concat}(y_a, y_b),
\end{aligned}
\tag{7}
$$

where $\text{s}()$ and $\text{b}()$ take both $x_a$ and $z$ as input. Specifically, each coupling layer includes three convolution layers where the first and last convolutions are $3 \times 3$, while the center convolution is $1 \times 1$. ELU (Clevert et al., 2015) is used as the activation function throughout the flow architecture:

$$
x \to \text{Conv}_{3\times 3} \to \text{ELU} \to \text{Conv}_{1\times 1} \oplus \text{FC}(z) \to \text{ELU} \to \text{Conv}_{3\times 3}
\tag{8}
$$

where $\text{FC}()$ refers to a linear full-connected layer and $\oplus$ is addition operation per channel between a 2D image and a 1D vector.

Importantly, $z$ is fed as conditional input to every coupling layers, unlike previous work (Agrawal & Dukkipati, 2016; Morrow & Chiu, 2019) where $z$ is only used to learn the mean and variance of the underlying Gaussian of $\upsilon$. This design is inspired by the generator in Style-GAN (Karras et al., 2019), where the style-vector is added to each block of the generator. We conduct experiments to show the importance of this architectural design (see §3.1).

**Refined Architecture of Glow.**    In this work, we refine the organization of these three elementary flows in one step (see Figure 3a) to reduce the total number of invertible $1 \times 1$ convolution flows. The reason is that the cost and the numerical stability of computing or differentiating the determinant of the weight matrix becomes the practical bottleneck when the channel dimension $c$ is considerably large for high-resolution images. To reduce the number of invertible $1 \times 1$ convolution flows while maintaining the permutation effect along the channel dimension, we use four split patterns for the $\text{split}()$ function in (7) (see Figure 3b). The splits perform on the channel dimension with continuous and alternate patterns, respectively. For each pattern of the split, we alternate $x_a$ and $x_b$. Coupling layers with different split types alternate in one step of our flow, as illustrated in Figure 3a. We further replace the original multi-scale architecture with the fine-grained multi-scale architecture (Figure 3c) proposed in Ma et al. (2019a), with the same value of $M = 4$. Experimental improvements over Glow demonstrate the effectiveness of our refined architecture (§3.1).

## 2.4 Tackling the Two Porblems in VAEs and Generative Flows

**Resolving Local Dependency in Generative Flows with Global Information from $z$.**    As discussed in §1.3, the flow-based decoder suffers the limitation of expressiveness and local dependency.

Table 1: Density estimation performance on four benchmark datasets. Results are reported in *bits/dim*.

| Model | CIFAR-10 8-bit | ImageNet 8-bit | LSUN-bedroom 5-bit | LSUN-bedroom 8-bit | CelebA-HQ 5-bit | CelebA-HQ 8-bit |
|---|---|---|---|---|---|---|
| **Autoregressive models** | | | | | | |
| IAF VAE (Kingma et al., 2016) | 3.11 | — | — | — | — | — |
| PixelRNN (Oord et al., 2016) | 3.00 | 3.63 | — | — | — | — |
| MAE (Ma et al., 2019b) | 2.95 | – | — | — | — | — |
| PixelCNN++ (Salimans et al., 2017) | 2.92 | – | — | — | — | — |
| PixelSNAIL (Chen et al., 2017b) | **2.85** | – | | | | |
| SPN (Menick & Kalchbrenner, 2019) | — | **3.52** | — | — | **0.61** | — |
| **Flow-based models** | | | | | | |
| Real NVP (Dinh et al., 2016) | 3.49 | 3.98 | — | — | — | — |
| Glow (Kingma & Dhariwal, 2018) | 3.35 | 3.81 | 1.20 | — | 1.03 | — |
| Glow: refined | 3.33 | 3.77 | 1.19 | 1.98 | 1.02 | 1.99 |
| Flow++ (Ho et al., 2019) | 3.29 | – | — | — | — | — |
| Residual Flow (Chen et al., 2019) | 3.28 | 3.76 | — | — | 0.99 | — |
| MaCow (Ma et al., 2019a) | 3.28 | 3.75 | 1.16 | — | **0.95** | — |
| **Our model** | **3.27** | **3.72** | **1.14** | **1.92** | 0.97 | **1.97** |

In the VAE framework of our model, the latent codes $z$ provides the decoder with the imperative global information, which is essential to resolve the limitation of expressiveness due to local dependency. On the other hand, the flow-based decoder favors to store local dependencies, encouraging the encoder to extract global information that is complementary to it.

**Resolving Posterior Collapse in VAEs with Flow-based Decoders.**  As discussed in previous work (Chen et al., 2017a), one possible reason for posterior collapse in VAEs is that the decoder model is sufficiently expressive such that it completely ignores latent variables $z$, and a solution to posterior collapse is to limit the capacity of the decoder. This suggests generative flow an ideal model for the decoder since they are insufficiently powerful to trigger the posterior collapse problem.

**Architectural Inductive Biases for Decoupled Representation Learning.**  From the high-level view of our model, we utilize these complementary properties of the architectures of the encoder and decoder as inductive bias to attempt to decouple the global and local information of an image by storing them in separate representations. The (indirect) supervision of learning global latent representation $z$ comes from two sources of architectural inductive bias. First, the compression architecture, which takes a high-dimensional image as input and outputs a low-dimensional vector, encourages the encoder to discard local dependencies of the image. Second, the preference of the flow-based decoder for capturing local dependencies reinforces global information modeling of the encoder, since the all the information of the input image $x$ needs to be preserved by $z$ and $v$.

## 3 EXPERIMENTS

To evaluate our generative model, we conduct two groups of experiments on four benchmark datasets that are commonly used to evaluate deep generative models: CIFAR-10 (Krizhevsky & Hinton, 2009), $64 \times 64$ downsampled version ImageNet (Oord et al., 2016), the *bedroom* category in LSUN (Yu et al., 2015) and the CelebA-HQ dataset (Karras et al., 2018)[2]. Unlike previous studies which performed experiments on 5-bit images from the LSUN and CelebA-HQ datasets, all the samples from the four datasets are 8-bit images in our experiments. All the models are trained by using affine coupling layers and uniform dequantization (Uria et al., 2013). Additional details on datasets, model architectures, and results of the conducted experiments are provided in Appendix C.

### 3.1 GENERATIVE MODELING

We begin our experiments with an evaluation on the performance of generative modeling, leaving the experiments of evaluating the quality of the decoupled global and local representations to §3.2.

---

[2]For LSUN datasets, we use $128 \times 128$ downsampled version, and for CelebA-HQ we use $256 \times 256$ version.

| Model | FID |
|-------|-----|
| PixelCNN[†] | 65.93 |
| PixelIQN[†] | 49.46 |
| DCGAN[‡] | 37.11 |
| WGAN-GP[‡] | 29.30 |
| EBM | 40.58 |
| NCSN | 25.32 |
| Glow | 46.90 |
| Glow: refined | 46.50 |
| Residual Flow | 46.37 |
| **Our model** | **37.52** |

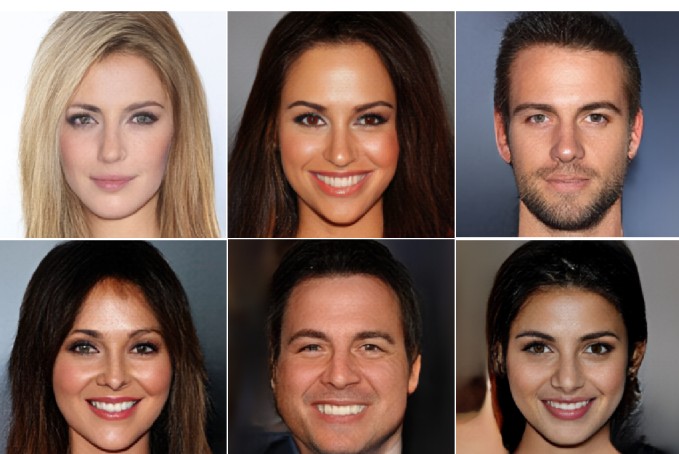

Table 2: FID scores on CIFAR-10.     Figure 4: 8-bit CelebA-HQ samples with temperature 0.7.

The baseline model we compare with is the refined Glow model, which is the exact architecture used in our flow-based decoder, except the conditional input $z$. Thus, the comparison with this baseline illustrates the effect of the decoupled representations on image generation. For the refined Glow model, we adjust the number of steps in each level so that there are similar numbers of coupling layers and parameters with the original Glow model for a fair comparison.

**Density Estimation.** Table 1 provides the negative log-likelihood scores in bits/dim (BPD) on the four benchmark datasets, along with the top-performing autoregressive models and flow-based generative models. For a comprehensive comparison, we report results on 5-bit images from the LSUN and CelebA-HQ datasets with additive coupling layers. Our refined Glow model obtains better performance than the original one in Kingma & Dhariwal (2018), demonstrating the effectiveness of the refined architecture. The proposed generative model achieves state-of-the-art BPD on all the four standard benchmarks in the non-autoregressive category, except the 5-bit CelebA-HQ dataset.

**Sample Quality** For quantitative evaluation of sample quality, we report the Fréchet Inception Distance (FID) (Heusel et al., 2017) on CIFAR-10 in Table 2. Results marked with † and ‡ are taken from [†]Ostrovski et al. (2018) and [‡]Heusel et al. (2017), respectively. Table 2 also provides scores of two energy-based models, EBM (Du & Mordatch, 2019) and NCSN (Song & Ermon, 2019). We see that our model obtains better FID scores than all the other explicit density models. In particular, the improvement over the refined Glow model on FID score demonstrates that learning decoupled representations is also helpful for realistic image synthesis.

Qualitatively, Figure 4 showcases some random samples for 8-bit CelebA-HQ $256 \times 256$ at temperature 0.7. More image samples, including samples on other datasets, are provided in Appendix F.

**Effect of feeding $z$ to every coupling layer.** As mentioned in §2.3, we feed latent codes $z$ to every coupling layer in the flow-based decoder. To investigate the importance of this design, we perform experiments on CIFAR-10 to compare our model with the baseline model where $z$ is only used in the underlying Gaussian of $\upsilon$ (Agrawal & Dukkipati, 2016; Morrow & Chiu, 2019). Table 3 gives the performance on BPD and FID score. Our model outperforms the baseline on both the two metrics, demonstrating the effectiveness of this design in our decoder.

Table 3: BPD and FID score.

| Model | BPD | FID |
|-------|-----|-----|
| Baseline | 3.31 | 43.34 |
| Ours | **3.27** | **37.52** |

### 3.2 DECOUPLED REPRESENTATION LEARNING

**Image Classification** As discussed above, good latent representation $z$ need to capture global features that characterize the entire image, and disentangle the underlying causal factors. From this perspective, we follow the widely adopted *downstream linear evaluation protocol* (Oord et al., 2018; Hjelm et al., 2019) to train a linear classifier for image classification on the learned representations using all available training labels. The classification accuracy is a measure of the linear separability,

| Model | Acc. |
|---|---|
| Raw pixel | 35.32 |
| AAE[†] | 37.76 |
| VAE[†] | 39.59 |
| BiGAN[†] | 44.90 |
| Deep InfoMax[‡] | 49.62 |
| **Our** ($z$) | 59.53 |
| **Our** ($v$) | 17.16 |

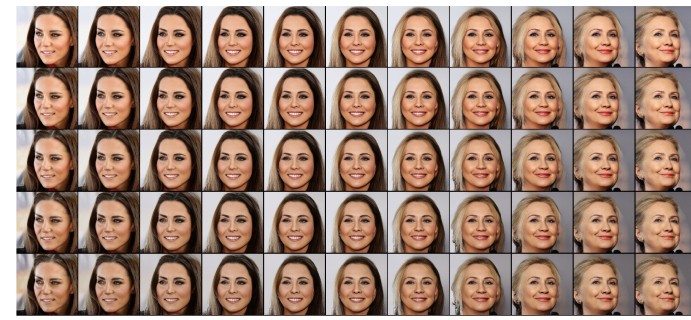

Table 4: Classification accuracy.  Figure 5: 2-dimensional linear interpolation between real images.

which is commonly used as a proxy for disentanglement and mutual information between representations and class labels. We perform linear classification on CIFAR-10 using a support vector machine (SVM). Table 4 lists the classification accuracy of SVM on the representations of $z$ and $v$, together with AAE (Makhzani et al., 2015), VAE (Kingma & Welling, 2014), BiGAN (Donahue et al., 2017) and Deep InfoMax (Hjelm et al., 2019). Results marked with † are token from Hjelm et al. (2019). Raw pixel is the baseline that directly training a classifier on the raw pixels of an image. The classification accuracy on the representation $z$ is significantly better than that on $v$, indicating that $z$ captures more global information, while $v$ captures more local dependencies. Moreover, the accuracy of $z$ outperforms Deep InfoMax, which is one of the state-of-the-art unsupervised representation learning methods via mutual information maximization.

**Two-dimensional Interpolation**   Our generative model leads to the *two-dimensional interpolation*, where we linearly interpolate the two latent spaces $z$ and $v$ between two real images:

$$
\begin{array}{rcl}
h(z) & = & (1-\alpha)z_1 + \alpha z_2 \\
h(v) & = & (1-\beta)v_1 + \beta v_2
\end{array}
\tag{9}
$$

where $\alpha, \beta \in [0,1]$. $z_1, v_1$ and $z_2, v_2$ are the global and local representations of images $x_1$ and $x_2$, respectively. Figure 5 shows one interpolation example from CelebA-HQ, where the images on the left top and right bottom corners are the real images[3]. The switch operation is two special cases of the two-dimensional interpolation with $(\alpha = 1, \beta = 0)$ and $(\alpha = 0, \beta = 1)$. More examples of interpolation and switch operation are provided in Appendix D.

**Discussion**   The results on image classification and two-dimensional interpolation empirical demonstrate that our model relive the posterior collapse problem in VAEs, by learning meaningful information in the latent space. Due to the entirely unsupervised setting, however, there are no intuitive explanations or theoretical guarantees for what information is captured by the global and local representations, respectively. It is an interesting direction for future work to explore how to learn more interpretable representations by investigating connections to architectural inductive biases, or leveraging weak or distant supervision.

## 4   RELATED WORK

**Combination of VAEs and Generative Flows.**   In the literature of combining VAEs and generative flows, one direction of research is to use generative flows as an inference machine in variational inference for continuous latent variable models (Kingma et al., 2016; Van Den Berg et al., 2018). Another direction is to incorporate generative flows in the VAE framework as a trainable component, such as the prior (Chen et al., 2017a) or the decoder (Agrawal & Dukkipati, 2016; Morrow & Chiu, 2019; Mahajan et al., 2019). Recently, two contemporaneous work (Huang et al., 2020; Chen et al., 2020) explore the idea of constructing an invertible flow-based model on an augmented input space by augmenting the original data with an additional random variable. The main difference between these work and ours is the purpose of introducing the latent variables and using generative flows. In

---

[3]For each column, $\alpha$ ranges in $[0.0, 0.25, 0.5, 0.75, 1.0]$; while for each raw, $\beta$ ranges in $[0.0, 0.1, 0.2, 0.3, 0.4, 0.5, 0.6, 0.7, 0.8, 0.9, 1.0]$

Huang et al. (2020); Chen et al. (2020), the latent variables are utilized to augment the input with extra dimensions to improve the expressivenss of the bijective mapping in generative flows. Our generative model, on the other hand, aims to learn representations with decoupled information, and the design of the latent variables and the flow-based decoder architecture is to accomplish this goal. The proposed generative model is closely related with generative flows with picecewise invertible transformations, such as RAD (Dinh et al., 2019) and CIFs (Cornish et al., 2020), and can be regarded as infinite mixtures of flows (Papamakarios et al., 2019).

**Disentangled Representation Learning.** Disentanglement learning (Bengio et al., 2013; Mathieu et al., 2016) recently becomes a popular topic in representation learning. Creating representations where each dimension is independent and corresponds to a particular attribute have been explored in several approaches, including VAE variants (Alemi et al., 2017; Higgins et al., 2017; Kim & Mnih, 2018; Chen et al., 2018; Mathieu et al., 2019), adversarial training (Mathieu et al., 2016; Karras et al., 2019) and mutual information maximization/regularization (Chen et al., 2016; Hjelm et al., 2019; Sanchez et al., 2019). Of particular relevance to this work are approaches that explore disentanglement in the context of VAEs, which aim to achieve independence or generalized decomposition (Mathieu et al., 2016) of the latent space. Different from these work which attempted to learn factorial representations for disentanglement, we aim to learn two separate representations to decouple the global and local information.

**Neural Style Transfer.** From the visualization, the switch operation of our model is also (empirically) related to neural style transfer of natural images (Gatys et al., 2015; Johnson et al., 2016; Jing et al., 2019) — the global and local representations in our model appear to correspond to the *style* and *content* representations in neural style transfer models. The main difference is that the global and local representations in our model are learned in unsupervised manner, while the content and style representations in neural style transfer models are usually extracted from a pre-trained classification network (VGG network) (Simonyan & Zisserman, 2014). It is an interesting direction of future work to further investigate the relation between our learned decoupled representations and the content and style representations in neural style transfer models.

## 5 CONCLUSION

In this paper, we propose a simple and effective generative model that embeds a generative flow as decoder in the VAE framework. Simple as it appears to be, our model is capable of automatically decoupling global and local representations of images in an entirely unsupervised setting. Experimental results on standard image benchmarks demonstrate the effectiveness of our model on generative modeling and representation learning. Importantly, we demonstrate the feasibility of decoupled representation learning via the plain likelihood-based generation, using only architectural inductive biases. Moreover, the two-dimensional interpolation supported by our model, with the switch operation as a special case, is an important step towards controllable image manipulation.

## ACKNOWLEDGMENTS

The authors would also like to thank Xiangyu Yue and Chunting Zhou for their helpful discussions during drafting this paper.

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

APPENDIX: DECOUPLING GLOBAL AND LOCAL REPRESENTATIONS VIA INVERTIBLE GENERATIVE FLOWS

## A   PRELIMINARY INTRODUCTION OF GLOW

Flow-based generative models focus on certain types of transformations $f_\theta$ that allow (i) the inverse functions $g_\theta$ and Jacobian determinants to be tractable and efficient to compute and (ii) $f_\theta$ to be expressive. Most work within this line of research is dedicated to designing invertible transformations to enhance the expressiveness while maintaining the computational efficiency (Kingma & Dhariwal, 2018; Ma et al., 2019a; Zhou et al., 2019; Chen et al., 2019; Ho et al., 2019), among which Glow (Kingma & Dhariwal, 2018) has stood out for its simplicity and effectiveness. The following briefly describes the three types of transformations that comprise Glow, which (in a refined version) is adopted as the backbone architecture of the flow-based decoder in our generative model (detailed in Appendix B).

**Actnorm.**   Kingma & Dhariwal (2018) proposed an activation normalization layer (Actnorm) as an alternative for batch normalization (Ioffe & Szegedy, 2015) to alleviate the challenges in model training. Similar to batch normalization, Actnorm performs an affine transformation of the activations using a scale and bias parameter per channel for 2D images, such that

$$y_{i,j} = s \odot x_{i,j} + b, \tag{1}$$

where both $x$ and $y$ are in shape $[h \times w \times c]$ with spatial dimensions $(h, w)$ and channel dimension $c$.

**Invertible $1 \times 1$ convolution.**   To incorporate a permutation along the channel dimension, Glow includes a trainable invertible $1 \times 1$ convolution layer to generalize the permutation operation as:

$$y_{i,j} = W x_{i,j}, \tag{2}$$

where $W$ is the weight matrix with shape $c \times c$.

**Affine Coupling Layers.**   Following Dinh et al. (2016), Glow includes affine coupling layers in its architecture of:

$$\begin{aligned}
x_a, x_b &= \text{split}(x) \\
y_a &= x_a \\
y_b &= \text{s}(x_a) \odot x_b + \text{b}(x_a) \\
y &= \text{concat}(y_a, y_b),
\end{aligned} \tag{3}$$

where $\text{s}(x_a)$ and $\text{b}(x_a)$ are outputs of two neural networks with $x_a$ as input. The $\text{split}()$ and $\text{concat}()$ functions perform operations along the channel dimension.

## B   IMPLEMENTATION DETAILS

### B.1   COMPRESSION ENCODER

The encoder first compresses the input image of size $[h \times h \times c]$ to the low-resolution tensor of size $4 \times 4 \times c'$. Then, with a fully-connected layer, the encoder transforms the output tensor to a vector of dimension $d_z$. Concretely, to compress the high-resolution images to low-resolution tensors, the encoder consists of levels of ResNet blocks (He et al., 2016). At each level, there are two ResNet blocks with the same number of hidden units and strides 1 and 2, respectively. Thus, after each level the input is compressed to half of the spatial dimensions: from $h \times h$ to $\frac{h}{2} \times \frac{h}{2}$. ELU (Clevert et al., 2015) is used as the activation function throughout the encoder architecture.

### B.2   SCALE TERM IN AFFINE COUPLING LAYERS

To model the scale term $s$ in (7), a straight-forward way is to take the output of the neural network as the logarithm of $s$. Formally, let $u$ denote as the output from the neural network described in (8). Then we can compute $s$ by taking the exponential function: $s = \exp(u)$. In practice, however, we found this formulation leads to numerical issues in model training. In our implementation, we calculate $s$ in the following way:

$$s = \alpha \cdot \tanh(\frac{u}{2}) + 1$$

where the constant $\alpha \in (0, 1)$. In this formulation, we restrict $s$ in the range of $[1 - \alpha, 1 + \alpha]$. For ImageNet, we set $\alpha = 0.5$ while for other datasets we used $\alpha = 1.0$. In the experiments, we found this formulation not only improved the numerical stability but also achieved better performance on density estimation and FID scores.

### B.3 PRIOR DISTRIBUTION IN VAEs

In this work, the prior distribution $p_\theta(z)$ in VAE is modeled with a generative flow with architecture similar to Glow. The generative flow also consists of three elementary invertible transformations: actnorm, invertible linear layer and affine coupling layer. The actnorm and invertible linear layer is similar to those in Ma et al. (2019c), with the difference that we did not use the multi-head mechanism. The affine coupling layer is similar to the one in Glow, which applies the split function across the dimension $d_z$. The neural networks for the scale and bias terms in affine coupling layers are implemented with multi-layer perceptrons (MLP).

## C EXPERIMENTAL DETAILS

### C.1 PREPROCESSING

We used random horizontal flipping for CIFAR10, and CelebA-HQ 256. For CIFAR-10, we also used random cropping after reflection padding with 4 pixels. For LSUN 128, we first centre cropped the original image, then downsampled to size $128 \times 128$.

### C.2 OPTIMIZATION

Parameter optimization is performed with the Adam optimizer (Kingma & Ba, 2014) with $\beta = (0.9, 0.999)$ and $\epsilon = 1\mathrm{e} - 8$. Warmup training is applied to all the experiments: the learning rate linearly increases to the initial learning rate $1\mathrm{e} - 3$. Then we use exponential decay to decrease the learning rate with decay rate is $0.999997$.

### C.3 HYPER-PARAMETERS

Table 5: Hyper-parameters in our experiments.

| Dataset | batch size | latent dim $d_z$ | weight decay | # updates of warmup |
|---------|-----------|------------------|--------------|---------------------|
| CIFAR-10, $32 \times 32$ | 512 | 64 | $1e-6$ | 50 |
| ImageNet, $64 \times 64$ | 256 | 128 | $5e-4$ | 200 |
| LSUN, $128 \times 128$ | 256 | 256 | $5e-4$ | 200 |
| CelebA-HQ, $256 \times 256$ | 40 | 256 | $5e-4$ | 200 |

### C.4 COMPARISON OF MODEL SIZE AND TRAINING SPEED

Table 6 provides the number of parameters of different models on CIFAR-10, together with the corresponding training time over one epoch (measured on four Tesla V100 GPUs). With similar model size, our refined Glow model obtains better performance and faster speed, demonstrating the effectiveness and efficiency of the refined architecture. Due to introducing the encoder, our VAE-based model contains a little bit more parameters and the training speed is a little slower than the refined Glow. Same as training a standard VAE model, ELBO is used as the training objective.

Table 6: Model size and training speed on CIFAR10.

| | # params | time/epoch (s) | bits/dim |
|---|----------|----------------|----------|
| Glow | 46.2 M | 210.2 | 3.35 |
| Glow: refined | 46.5 M | 120.9 | 3.33 |
| Ours | 51.8 M | 144.8 | 3.27 |

Similar to VAE-based models, exact inference is not preserved and negative log-likelihood (NLL) is approximated with ELBO in model evaluation. However, since EBLO provides a way to estimate the upper bound of NLL, the scores in Table 1 are comparable to previous work.

# D    EXAMPLES FOR TWO-DIMENSIONAL INTERPOLATION

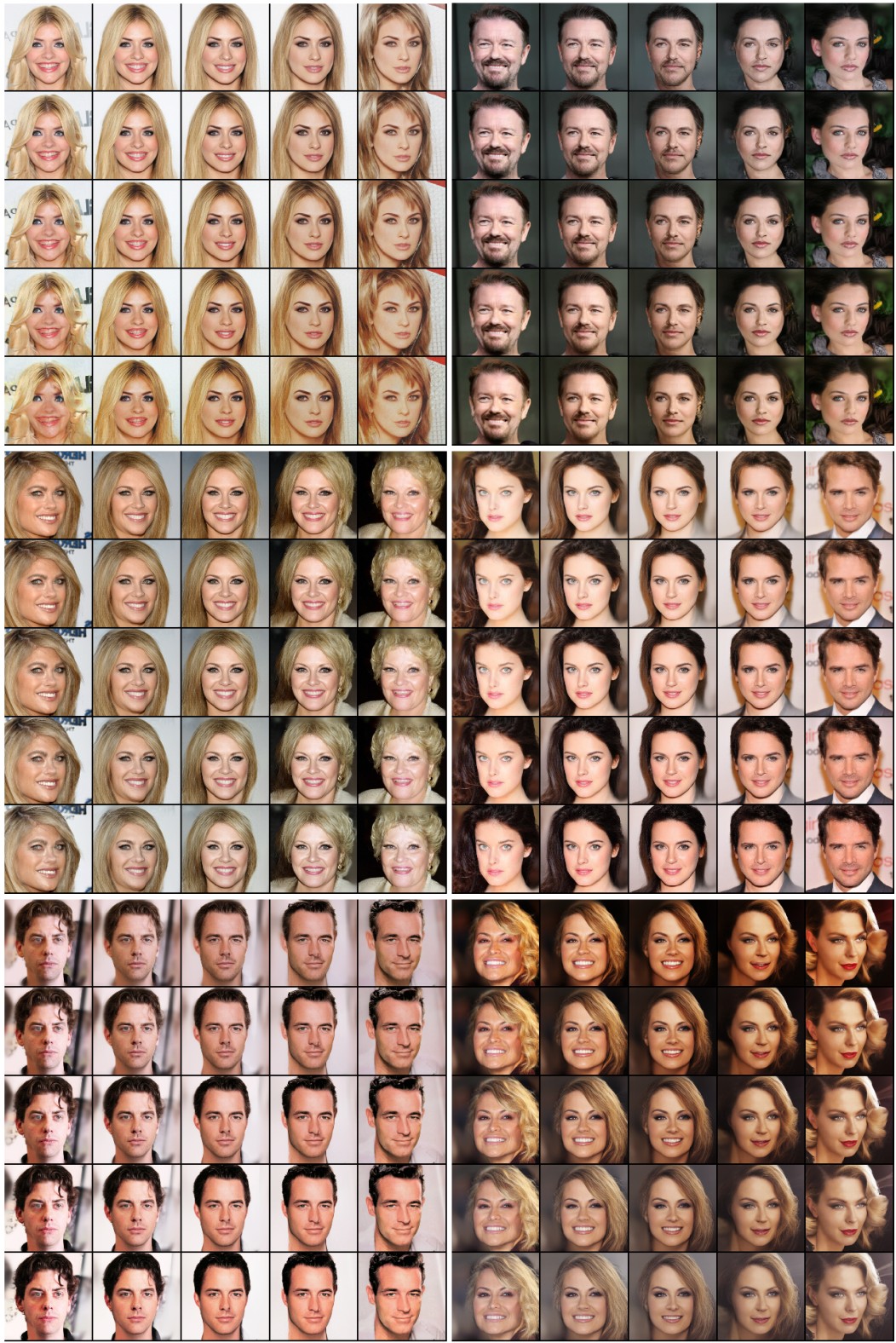

Figure 6: Interpolation operation between samples from 8-bit, 256×256 CelebA-HQ.

# E    MORE SAMPLES FOR SWITCH OPERATION

## E.1    CELEBA-HQ

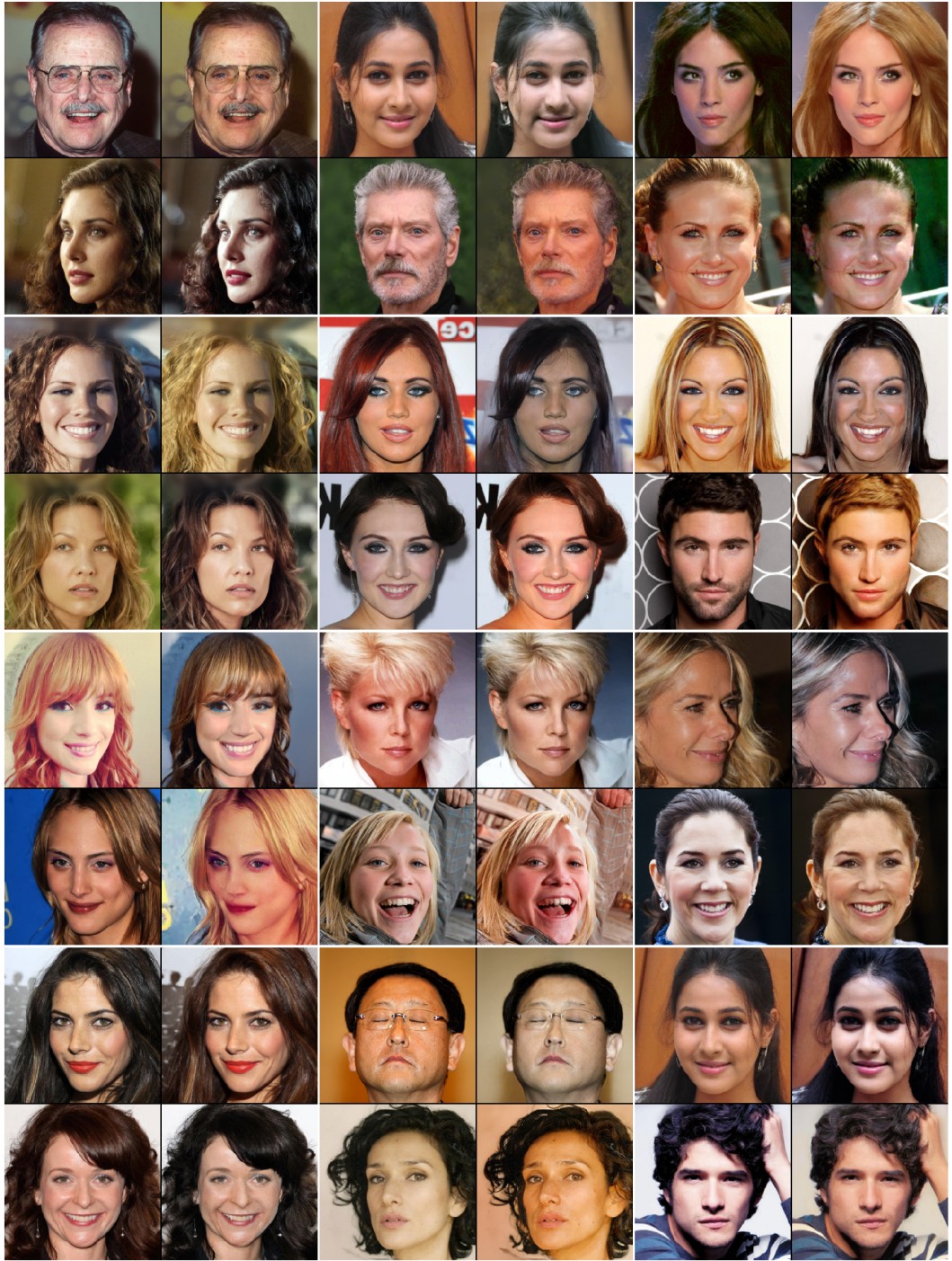

Figure 7: Switch operation between samples from 8-bit, 256×256 CelebA-HQ.

## E.2 CIFAR-10 & IMAGENET

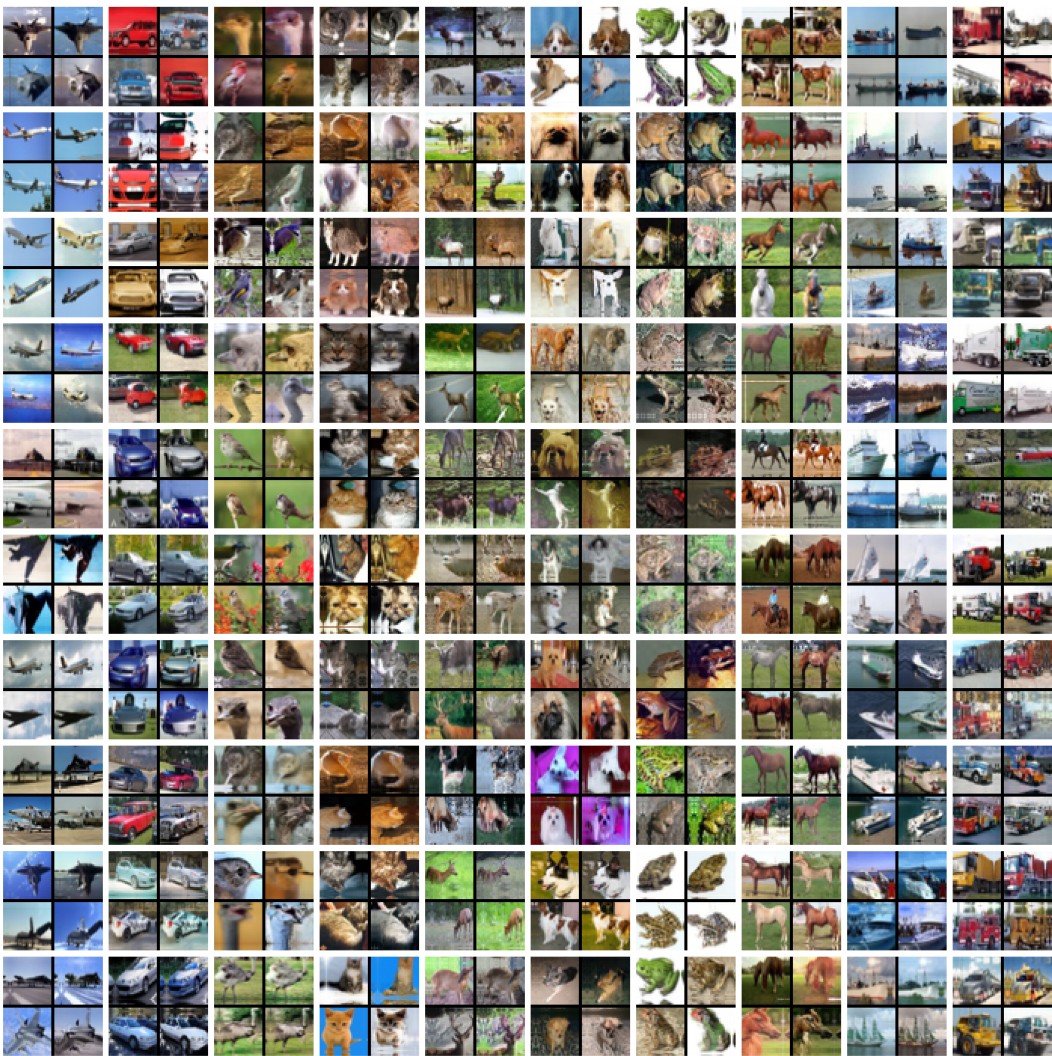

Figure 8: Switch operation between samples within the same class from 8-bit, 32×32 CIFAR-10.

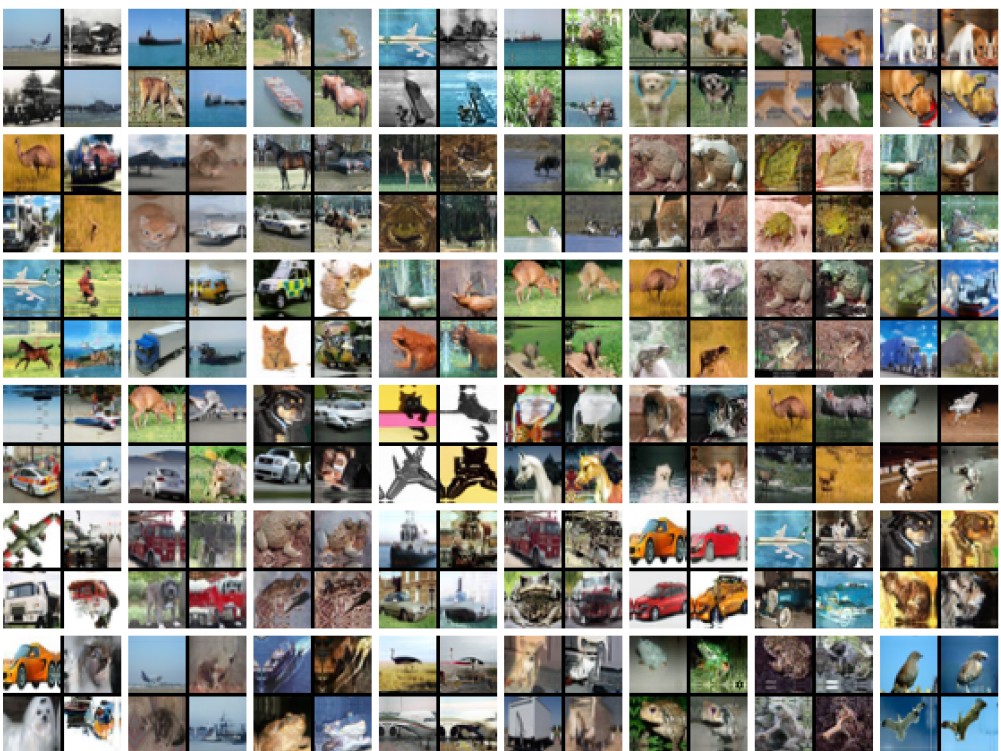

Figure 9: Switch operation between samples across different classes from 8-bit, 32×32 CIFAR-10.

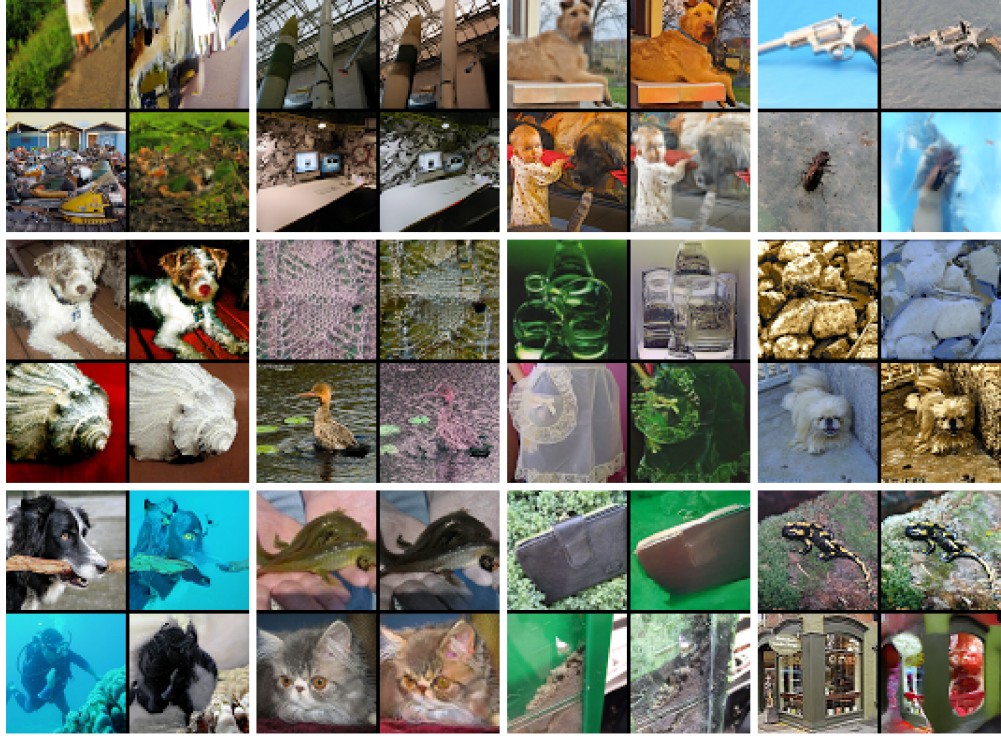

Figure 10: Switch operation between samples from 8-bit, 64×64 imagenet.

### E.3  LSUN-BEDROOM

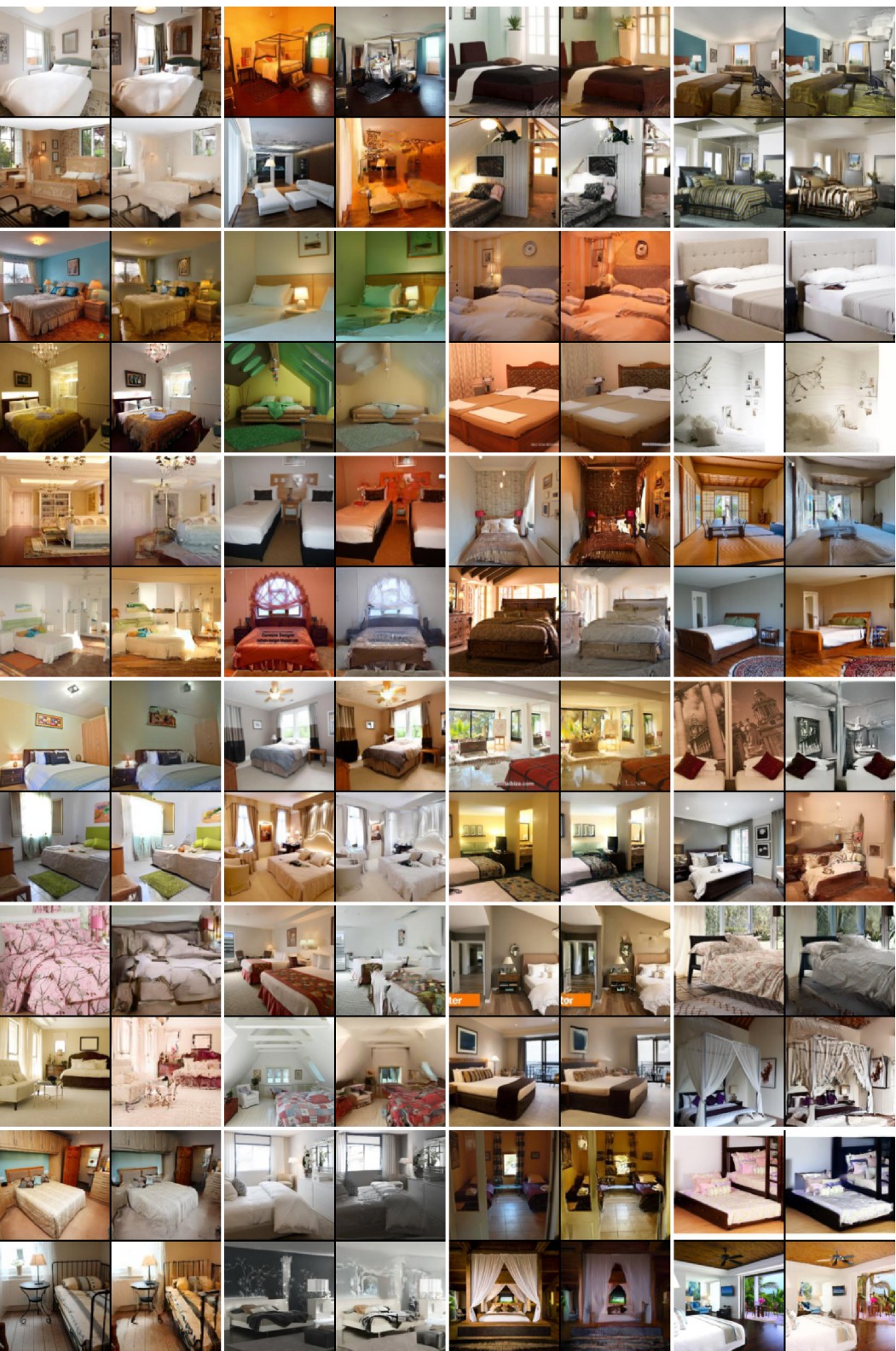

Figure 11: Switch operation between samples from 8-bit, 128×128 LSUN bedroom.

## F  MORE IMAGE SAMPLES

### F.1  CELEBA-HQ

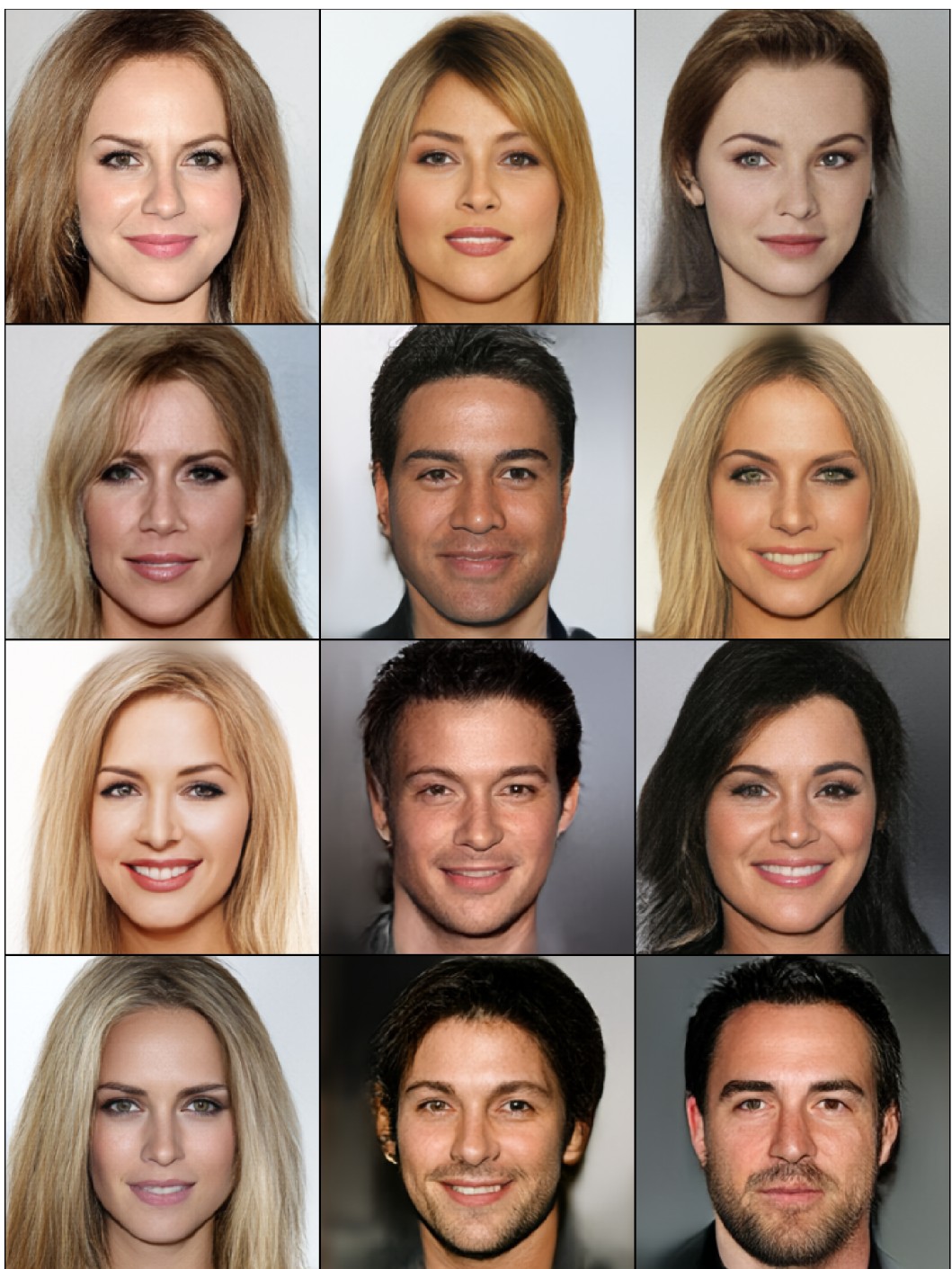

Figure 12: Samples from 8-bit, 256×256 CelebA-HQ with temperature 0.7.

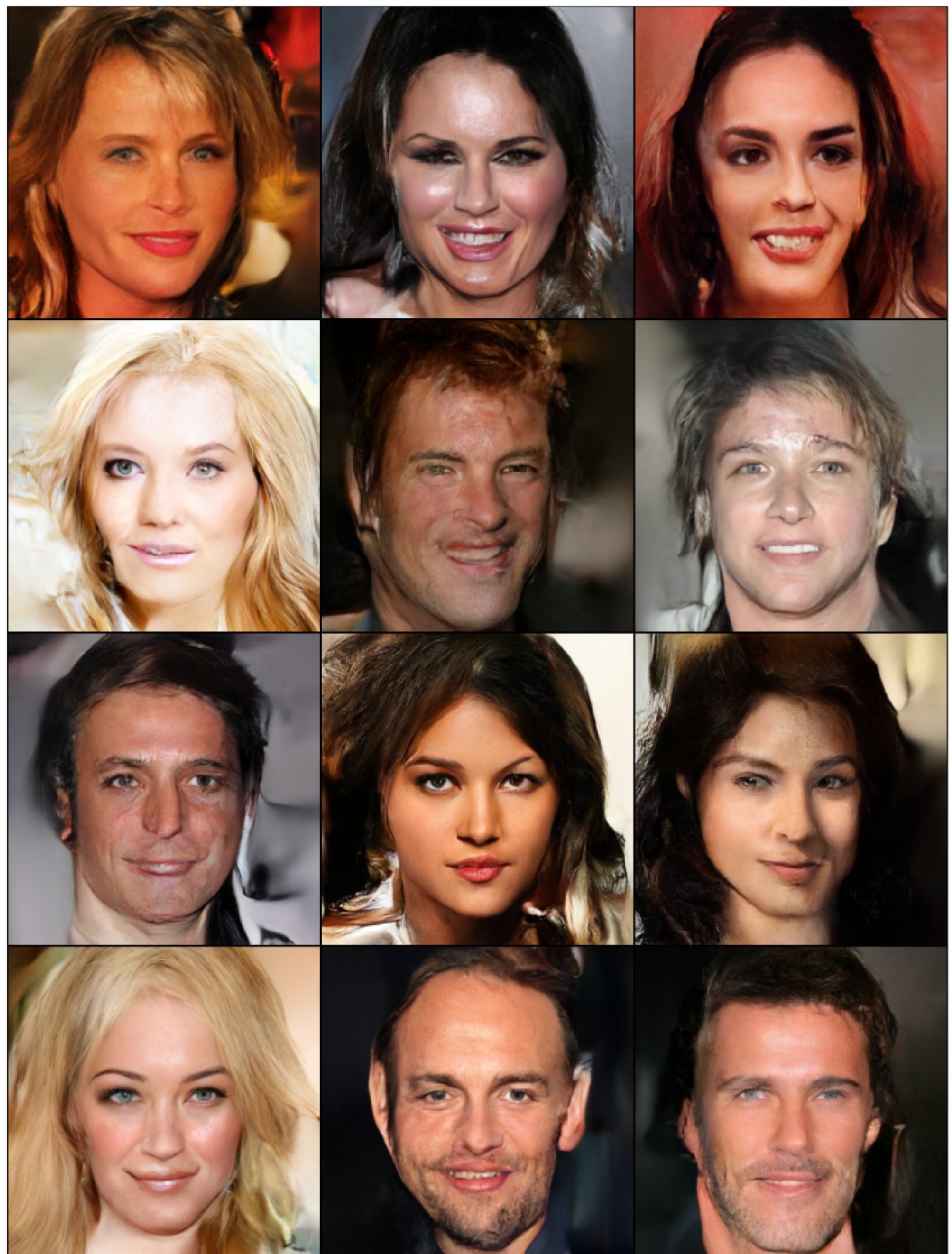

Figure 13: Samples from 8-bit, 256×256 CelebA-HQ with temperature 1.0.

## F.2 LSUN-BEDROOM

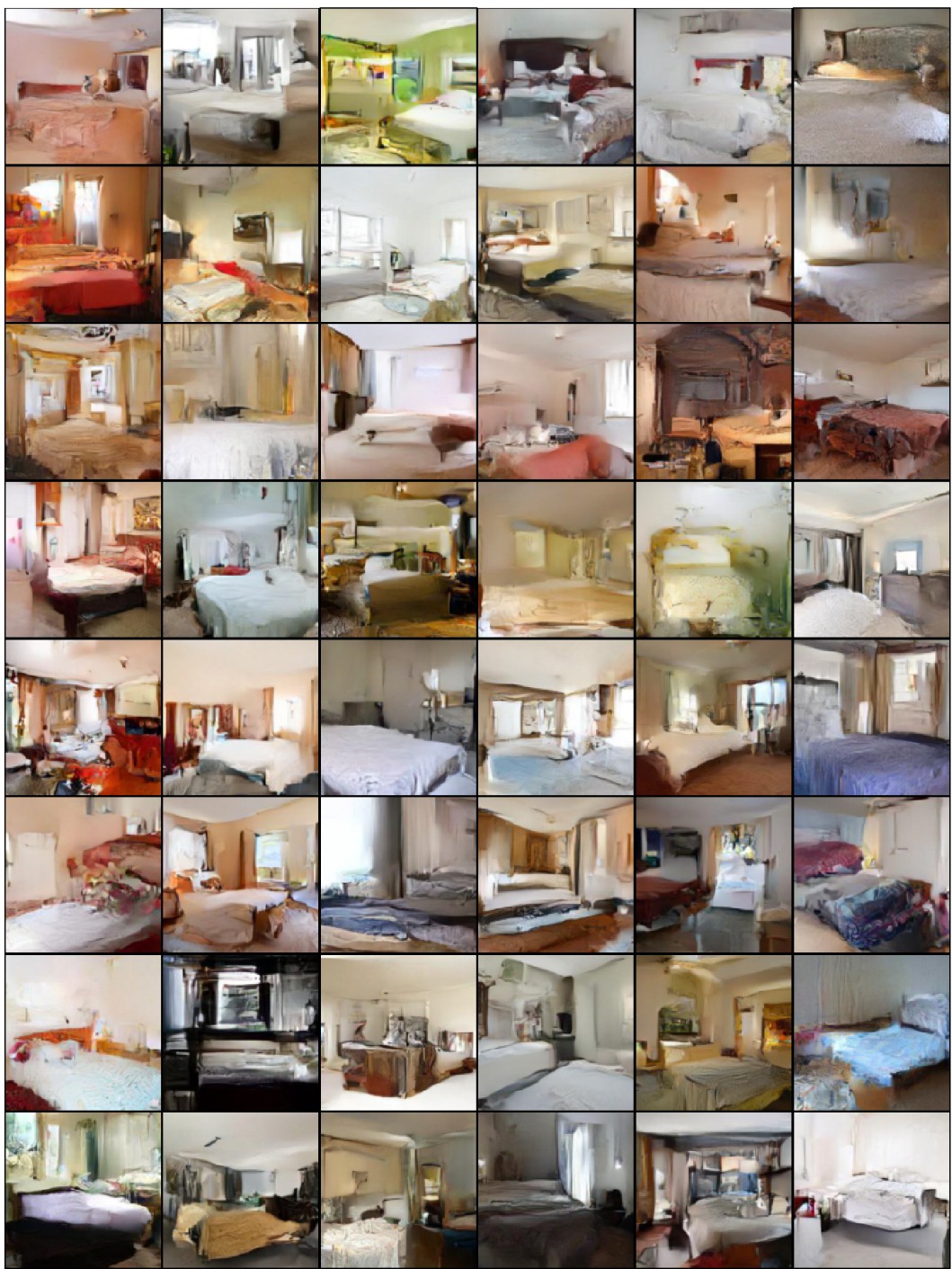

Figure 14: Samples from 8-bit, 128×128 LSUN bedrooms.

## F.3 CIFAR-10 & IMAGENET

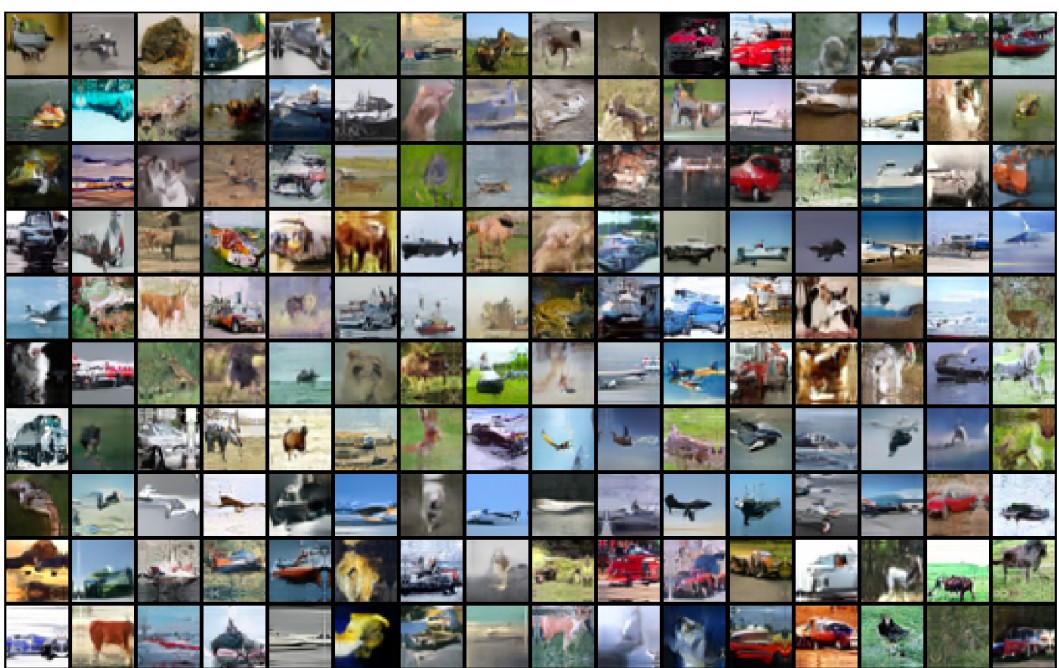

Figure 15: Samples from 8-bit, 32×32 CIFAR-10.

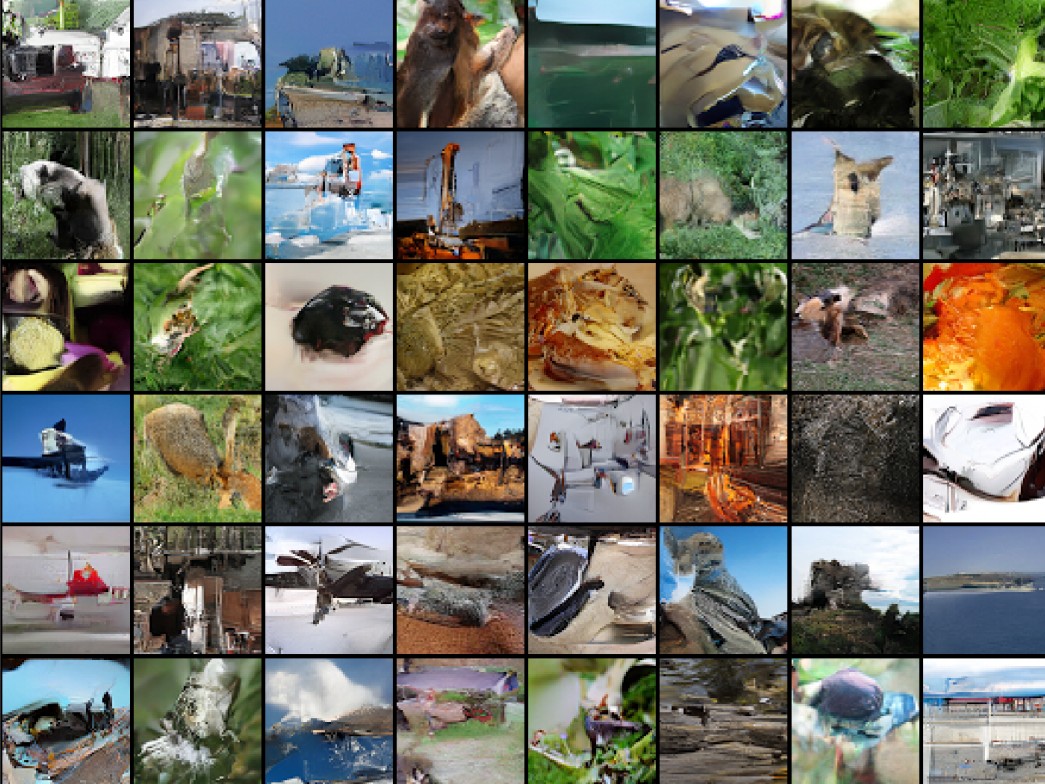

Figure 16: Samples from 8-bit, 64×64 imagenet.

