# OpenReview forum: "Decoupling Global and Local Representations via Invertible Generative Flows"
_ICLR.cc/2021/Conference — ICLR 2021 Poster_

### Official Review · AnonReviewer4 · 2020-10-27
**Interesting approach, but I'm not sure about the author claims. Are you really inferring global representations?**

**Rating:** 7
**Confidence:** 3

**Review:**

Pros:

-> I like the idea of conditional generative flows, where a low-dimensional embedding captures high-level features and a larger embedding latent space captures local representations.
-> Use a compression encoder to enforce informative low-dimensional embeddings
-> Good experimental results

Cons:

-> Can you really say that the low-dimensional embedding z is capturing GLOBAL variables? To me, global variables in a generative model drives non iid samples, hence correlating samples with each other. For instance, see the paper

Diane Bouchacourt, Ryota Tomioka, and Sebastian Nowozin. Multi-level variational autoencoder: Learning disentangled representations from grouped observations. In Thirty-Second AAAI Con- ference on Artificial Intelligence, 2018.

-> The author should clearly explain the training cost function. At the end they are combining a flow generative model (typically trained using ML) with an encoder VAE like posterior approximation. I guess they optimize some soft of ELBO, but it is not clear to me yet.

Overall, I like the paper and would like to see it accepted.

---

> ### Author Response · Authors · 2020-11-21
> **Response to review by AnonReviewer4**
>
> Thanks for your comments and positive feedback!
>
> We respond below to your questions and comments. We kindly request that you briefly read these responses and let us know if they do not fully satisfy your concerns.
>
> > Can you really say that the low-dimensional embedding z is capturing GLOBAL variables?
>
> From the two-dimensional interpolation experiments and the examples of the switch operation in Figure. 1, we see that the representation Z captures high-level features for the global properties of the image. And the results on image classification in Table 4 further justifies this observation.
>
> > The author should clearly explain the training cost function. At the end they are combining a flow generative model (typically trained using ML) with an encoder VAE like posterior approximation. I guess they optimize some soft of ELBO, but it is not clear to me yet.
>
> We appreciate your suggestion to clearly explain the training cost function. In our revised submission, we add section 2.1 to illustrate the high-level architecture of our generative model, where we clearly say that ELBO is the training objective function. The formula of EBLO is provided in Eq. (2).
>
> We hope the above explanation answers your questions. Thanks again for your positive feedback.

---

### Official Review · AnonReviewer2 · 2020-10-28
**Review for Decoupling Global and Local ....**

**Rating:** 7
**Confidence:** 4

**Review:**

The paper introduces a mixture of flows with the specific intent to disentangle global and local representations, to improve visual quality of samples. Architectures are borrowed from style-gan literature. The contributions are mainly architectural and empirical. The demonstrates improved visual quality over other normalizing flow methods.

Strengths
The paper introduces a straightforward latent variable model for flows which is designed in such a way that training on NLL gives good sample quality, which is measured in FID. Even further, the model also performs quite well on the NLL objective itself. Since the focus of normalizing flow literature has been mainly on NLL, I think this paper is a nice complement to existing literature.

Weaknesses:
- The paper never puts the equations from the background section together in the final objective. It would be helpful to have an equation representing the final objective with some of the relevant variables (i.e. log p(x) >= ... with variables x, z, and v).
- The paper does not introduce many novel theory or methods. This is not really a problem, but the paper can be better connected to existing work and clarified in this respect. The model the authors propose is very reminiscent of "infinite mixtures of flows" as outlined by (Papamakarios et al. "Normalizing Flows for Probabilistic Modeling and Inference" page 32). An example would be "Continuously Index Flows" by Cornish et al.. Note their method was introduced with a different intend in a different manner, so I think it would only make the paper better by citing these methods.
- Conditioning on a context variable in flow layers is not new (see Kingma et al., "Improved variational inference with inverse autoregressive flow."  and Lu et al. "Structured Output Learning with Conditional Generative Flows."). This is not really a problem, but again this should be clarified.
- How much added computation is required by the encoder model plus FCnet in eq. 7 compared to the Glow-refined model on which the proposed method is based?
- Perhaps the naming "compressing encoder" is not particularly useful. It implies a direct connection to actual image compression, which is as far as I understand not the case. Other than that, this seems like a fairly standard VAE encoder other than the size difference between x and z.

---

> ### Author Response · Authors · 2020-11-21
> **Response to review by AnonReviewer2**
>
> Thanks for your comments and positive feedback!
>
> We respond below to your questions and comments. We kindly request that you briefly read these responses and let us know if they do not fully satisfy your concerns.
>
> >Q1: The paper never puts the equations from the background section together in the final objective.  It would be helpful to have an equation representing the final objective with some of the relevant variables (i.e. log p(x) >= ... with variables x, z, and v)
>
> We appreciate your suggestion to clearly explain the training objective. In our revised submission, we have added section 2.1 to illustrate the high-level architecture of our generative model, where we clearly explain that ELBO is the training objective function. The formula of EBLO is provided in Eq. (2).
>
> >Q2: The model the authors propose is very reminiscent of "infinite mixtures of flows" as outlined by (Papamakarios et al. "Normalizing Flows for Probabilistic Modeling and Inference" page 32). An example would be "Continuously Index Flows" by Cornish et al. Conditioning on a context variable in flow layers is not new. This is not really a problem, but again this should be clarified.
>
> Thanks for pointing out the related work we missed. We have elaborated them in the related work section of the revised version.
>
> >Q3: How much added computation is required by the encoder model plus FCnet in eq. 7 compared to the Glow-refined model on which the proposed method is based?
>
> We have provided Table 6 in Appendix C.4 to show the comparison between different models on CIFAR-10, on the model size, the corresponding training time over one epoch (measured on four Tesla V100 GPUs), and the performance of density estimation (bits/dim). Due to introducing the encoder, our VAE-based model contains a little bit more parameters and the training speed is a little slower than the refined Glow.
>
> We hope the above explanation answers your questions. Thanks again for your positive feedback.

---

> > ### Comment · AnonReviewer2 · 2020-11-21
> > **Response**
> >
> > I would like to thank the authors for their response and clarifications. In large part I am satisfied with the response. In my opinion having the objective more explicitly (sec 2.1) in addition to the description in text would still be preferable. I have raised my score to 7 because of the clarifications.

---

> > > ### Author Response · Authors · 2020-11-24
> > > **Thanks for your feedback**
> > >
> > > We appreciate your swift feedback and your suggestion to explicitly add an equation of the objective function in Section 2.1.
> > > We have further revised the submission to reflect your suggestion.

---

### Official Review · AnonReviewer3 · 2020-10-28
**Interesting general ideas, but critical flaws in the presentation**

**Rating:** 6
**Confidence:** 4

**Review:**

##### Summary

This paper aims to improve a Normalizing Flow generative model, in particular
Glow, by conditioning the flow on global information of the image in the form
of a latent vector learned with the VAE framework. This latent vector is
injected at the scale and bias terms of the affine coupling layers of the flow
(inspired by what Style-GAN does at the batchnorm layers).

Unfortunately, critical aspects of the method remain unclear or unspecified. To
the best of my understanding, the paper lacks a clear explanation of the
complete pipeline used for training the method and the final objective
function. For evaluation, sampling and likelihood computation, procedures are
not completely specified either.

##### Pros
- The general ideas of the paper are well motivated. Combining the advantages of
  explicit likelihood and latent variable generative models is in my opinion an
  extremely interesting research direction.
- The authors propose architectural improvements to Glow and craft a conditional
  version that can effectively incorporate additional information to the flow.
- The authors model achieves improved or competitive density estimation,
  sampling, and downstream performance across various datasets, compared to the
  state-of-the-art.
- Some degree of local and global properties disentanglement is demonstrated in
  the qualitative results, showing the proposed direction is a promising one in
  that regard.

#####  Cons
The main drawback is in my view the presentation of the method. The method part
of the paper (Section 2) mixes theoretical justification with architecture
details and fails to clearly explain the full pipeline and training
objective. This made it very difficult if not impossible to analyze it and draw
conclusions. The second half of the paper is dedicated to experimental results,
yet the sampling procedure and likelihood computation are not clearly explained
either.

I think the submission would have been much stronger if the authors clearly
explained the whole method and dedicated more of the paper to a careful analysis
and justification of the design decisions and of some of the claims
(e.g. avoiding posterior collapse, global/local disentanglement).



Going into detail, by reading the abstract I get the impression that the VAE
framework is used, and the normalizing flow is used to model the generative
distribution $p(x|z)$. Yet the abstract also claims to only use a plain
log-likelihood objective as in explicit likelihood models, instead of the VAE
ELBO. Alternatively, I thought the latent code was learned with a separate VAE,
but the introduction states that the generative flow is "[embedded] in the VAE
framework to model the decoder".

Assuming that the VAE framework is used with a (conditional) normalizing flow
for decoder, Section 2 introduces more confusion when the authors state "we feed
$z$ as a conditional input to a flow-based decoder, which transforms $x$ into
the representation $\nu$ with the same dimension." This is confusing since
typically the decoder input should be a low-dimensional representation, but here
it seems to be the image as well, so the concept of decoding seems ill
placed. Moreover, if this is the case, what would be the point of the
reconstruction term in the ELBO if invertibility guarantees perfect
reconstruction? Shouldn't the flow output $\nu$ be a stochastic latent code as
in VAEs? Is a prior density regularization imposed on $\nu$ as well?

Finally, another option would be that everything is trained with the negative
log-likelihood cost of the normalizing flow. This would be consistent with the
claim in the abstract that only a plain log-likelihood is utilized. But in that
case, what is the justification for using a stochastic encoder if it is not
regularized? How can it be guaranteed that $z$ will not be ignored? What is the
role of $z$ during sampling? Does the likelihood computation involve $z$ or just
$\nu$?

I apologize for writing my internal thought process but I also wanted to convey
that even if the method was clarified to be some of the options I described, or
another one, many of the design decisions would still require extra justification
and analysis.


Other comments:
- Although Figure 1 shows capturing of some global color properties, looking at
  the interpolations in Figure 5, there seems to be little variability
  w.r.t. $z$, so maybe the claims of learning "decoupled" and "disentangled"
  representations require more justification.

- About the initialization of the weights of the last linear layer with zeroes
  (Section 2.1). Wouldn't this create a null output at thus a null backward
  gradient in the first iteration? Even if a non-zero bias was used, wouldn't an
  unbreakable symmetry condition be produced?


************
After Rebuttal:

I thank the authors for their multiple clarifications, and apologize for my initial misunderstanding.
I understand now that the flow model is used to compute $p(x|z)$ as a function of $x$ and $z$. Maybe the "decoder" terminology is a bit confusing here, but this is quite a nice idea overall. It would have been nice to see multiple samples from  $p(x|z)$ for a fixed $z$, to evaluate the expressiveness of the model.

I'm raising my score to acceptance.

PS: Some typos remain in the revised version, eg: "varnishes", "tne"

---

> ### Author Response · Authors · 2020-11-21
> **Response to review by AnonReviewer3**
>
> Thanks for your time and constructive comments! We appreciate your positive feedback on the good motivation, interesting research direction, architectural improvements, effective methods, and promising performance quantitatively and qualitatively. We have made several modifications to improve the presentation of our work. Please refer to the revision summary for more details: https://openreview.net/forum?id=iWLByfvUhN&noteId=Qr5bIoltAES
>
> We respond below to your questions and comments. We kindly request that you briefly read these responses and let us know if they do not fully satisfy your concerns.
>
> >Q1: The main drawback is in my view the presentation of the method. The method part of the paper (Section 2) mixes theoretical justification with architecture details and fails to clearly explain the full pipeline and training objective. This made it very difficult if not impossible to analyze it and draw conclusions. The sampling procedure and likelihood computation are not clearly explained either.
>
> We appreciate your suggestion to clearly explain the full pipeline and training objective. In our revised submission, we add Section 2.1 to illustrate the high-level architecture of our generative model, where we clearly say that ELBO is the training objective function. The formula of EBLO is provided in Eq. (2).
>
> We also describe the generative process of our model in this section.
> More explanation of sampling procedure and likelihood computation are also provided in Section 2.1
>
> >Q2: I think the submission would have been much stronger if the authors clearly explained the whole method and dedicated more of the paper to a careful analysis and justification of the design decisions and of some of the claims (e.g. avoiding posterior collapse, global/local disentanglement).
>
> We appreciate your suggestion and have elaborated the section 2.4 to provide a high-level motivation of our model architecture design w.r.t tackling the two problems of posterior collapse in VAEs and local dependency in flows, and how the model decouples the global/local representations.
>
> >Q3: By reading the abstract I get the impression that the VAE framework is used, and the normalizing flow is used to model the generative distribution p(x|z). Yet the abstract also claims to only use a plain log-likelihood objective as in explicit likelihood models, instead of the VAE ELBO.
>
> We apologize for the confusion. We use the VAE framework with ELBO as the training objective.
> We have rewritten this into “likelihood-based objective” in the revised version to make it clearer.
>
> >Q4: Assuming that the VAE framework is used with a (conditional) normalizing flow for decoder, Section 2 introduces more confusion when the authors state "we feed z as a conditional input to a flow-based decoder, which transforms x into the representation ν with the same dimension."
>
> Thanks for your questions. We have rewritten section 2.1 to separately describe the training and generative processes of our model. We have also added more explanations to answer your questions and make this section clearer.
>
> Concretely, In the training process, we need to compute the distribution $p(x|z)$.
> Thus, X and Z are fed into the function $f$ of the flow-based decoder to compute the latent variables $\upsilon$ of the generative flow (see section 1.2 for the introduction of generative flows).
> In the generative process, we first sample $z$ and $\upsilon$ from their prior distributions (typically normal distribution). Then we input $z$ and $\upsilon$ into the inverse function $f^{-1}$ to generate an image $x = f_\theta^{-1}(\upsilon, z) = g_\theta(\upsilon, z)$.
>
> >Q5: I apologize for writing my internal thought process but I also wanted to convey that even if the method was clarified to be some of the options I described, or another one, many of the design decisions would still require extra justification and analysis.
>
> Thanks for sharing your internal thoughts. We would like to point out that almost all the questions are from the presentation of our work. We have carefully revised our paper to address your concerns and added more justification and analysis in the [revised version](https://openreview.net/forum?id=iWLByfvUhN&noteId=Qr5bIoltAES).
>
> > About the initialization of the weights of the last linear layer with zeroes (Section 2.1). Wouldn't this create a null output at thus a null backward gradient in the first iteration?
>
> Note that in practice, we model $\log \sigma^2(x)$ instead of directly modeling $\sigma^2(x)$. Thus, the zero-initialization produces a simple normal distribution for the posterior distribution at the beginning of training process, i.e. $\mu(x)=0$ and $\sigma^{2}(x) = 1$. This makes the training process much more stable at the first few updates.
>
> We hope the above explanation and our revised paper can address your concerns and facilitate the understanding of our paper. We sincerely appreciate it if further feedback could be provided.

---

> ### Author Response · Authors · 2020-11-25
> **Thanks for your feedback!**
>
> Thanks for your feedback. We have revised the submission to fix the typos you pointed out. We will add the experiments of multiple samples from the decoder for a fixed z in later revisions.

---

### Official Review · AnonReviewer1 · 2020-10-28
**Official Blind Review #1**

**Rating:** 8
**Confidence:** 4

**Review:**

Summary: The authors propose a novel combination of VAEs and Flow models, where the decoder is modelled through a conditional flow taking as input a “local” representation of the size of the input image and a “global” representation output by the encoder. The authors evaluate the proposed method on density estimation, quality of generations and linear probing on a variety of datasets and show improvement over state of the art.

Great:
* Conceptually simple method that seems to work quite well in practice, for this class of models.
* The linear probing experiment is quite convincing in justifying the use of “global” and “local” characterizations of the learned representations. So are the interpolations.

Could be improved:
* It’s not clear to what extent each of the proposed refinements to Glow (reorganization, different splits, fine-grained multi-scale architecture) improves Glow’s performance.

The authors propose a novel combination of known methods, evaluate it extensively and show considerable improvements over current state of the art. A clear accept.

---

> ### Author Response · Authors · 2020-11-21
> **Response to review by AnonReviewer1**
>
> Thanks for your comments and positive feedback!
>
> We respond below to your questions and comments. We kindly request that you briefly read these responses and let us know if they do not fully satisfy your concerns.
>
> > It’s not clear to what extent each of the proposed refinements to Glow (reorganization, different splits, fine-grained multi-scale architecture) improves Glow’s performance.
>
> We have provided Table 6 in Appendix C.4 to show the comparison between the original and refined Glow in terms of model size, training speed, and the performance of density estimation. From the table, we see that the refined Glow (2nd row) has a similar number of parameters, while the training speed is much faster, and the performance of density estimation (bits/dim) is better than the original Glow (1st row). Based on our observations, the speedup of the refined Glow mainly comes from reducing the number of the invertible 1x1 convolution flows by re-organization and different splits, and the better bits/dim performances mainly come from the fine-grained multi-scale architecture. We will provide more analysis in the final version.
>
> We hope the above explanation answers your questions. Thanks again for your positive feedback.

---

### Author Response · Authors · 2020-11-21
**Common Remarks and Summary of Revision v1**

We thank all the reviewers for their insightful and positive feedback. We are especially encouraged that they found the problem we focus on to be important and interesting (R3 & R4), and our motivation and idea to be clear, strong, and novel (R3 & R4). We are glad they found our approach to be effective and novel (R1, R2, R3 & R4), evaluated with extensive experiments (R1 & R3), convincing experiments that can justify the proposed model (R1 & R3), and achieving superior results (R1, R2, R3 & R4).

We summarize the revision as follows:

1. Adding section 2.1 to illustrate the general framework of our generative model, including the training objective and the generative process. More explanation of Sampling procedure and likelihood computation are also provided.

2. Revising the section 2.4 to provide a high-level motivation of our model architecture design w.r.t tackling the two problems of posterior collapse in VAEs and local dependency in flows, and how the model decouples the global/local representations.

3. Adding more explanation of the method and more analysis and justification of the design and claims.

4. Elaborating the section 4 of related work.

5. Fixing typos and writing issues.

---

### Decision · Program_Chairs · 2021-01-07
**Final Decision**

**Decision:**

Accept (Poster)

**Comment:**

The paper proposes a hybrid VAE-normalizing-flow for extracting local and global representations of images.  While the reviewers found the model itself to be "conceptually simple" and "straightforward", all were convinced by the empirical evaluation that, indeed, interesting representation learning is going on, resulting in a unanimous vote to accept.